# The Emergencies in the Group of Patients with Temporomandibular Disorders

**DOI:** 10.3390/jcm12010298

**Published:** 2022-12-30

**Authors:** Malgorzata Pihut, Malgorzata Kulesa-Mrowiecka

**Affiliations:** 1Jagiellonian University Medical College, Prosthodontic and Orthodontic Department, Institute of Dentistry, 4 Montelupich Str., 31-155 Krakow, Poland; 2Jagiellonian University Medical College, Department of Rehabilitation in Internal Diseases, Faculty of Health Science, Institute of Physiotherapy, 8 Skawińska Str., 31-066 Krakow, Poland

**Keywords:** emergencies, temporomandibular disorders, TMD, management of TMD, myospasm of pterygoid muscle, rehabilitation, differential diagnosis, subluxation of mandible

## Abstract

Temporomandibular disorder is a musculoskeletal disease with complex, multifactorial etiology regarding improper functioning of the stomatognathic system (masticatory muscles, temporomandibular joints, and surrounding structures). This article presents medical emergencies occurring among patients treated for temporomandibular disorders, which tend to constitute a severe difficulty for practitioners during their clinical practice. Examples of the most common emergencies of this type are disc displacement without reduction and a sudden contraction of the inferior part of the lateral pterygoid muscle. The latter occurs in cases of uncontrolled and incorrect use of the anterior repositioning splints and the hypertrophy of the coronoid process of the mandible. The sudden attacks of pain of secondary trigeminal neuralgia are also discussed in this article, together with their specific nature, which is significantly different from the nature of the pain of primary trigeminal neuralgia, yet the two types of neuralgia can be easily confused when the primary one takes the painful form. Subsequent emergencies discussed are myofascial pain syndrome, traumatic and inflammatory states of the temporomandibular joints, subluxation, and the consequences of intense occlusive parafunctions. Finally, the recommended therapeutic methods, which are used as part of the treatment in the cases of aforementioned emergencies, are described in this mini-review article, emphasizing that the implementation of the incorrect treatment and rehabilitation for emergencies of temporomandibular disorders may lead to permanent damage to the soft tissue structures of the temporomandibular joints.

## 1. Introduction

Temporomandibular disorders (TMD) constitute an increasingly common disorder that affects the functioning of the masticatory muscles, temporomandibular joints, and surrounding structures in the stomatognathic system. TMD most often affects people in their third and fourth decade of life; however, nowadays, adolescents are increasingly susceptible to such health conditions [1,2,3]. Moreover, the increase in the occurrence of significant pain in the masticatory system is observed, which results in hindering everyday functioning and the performance of professional duties. One of the main etiological factors is excessive stress causing occlusal parafunctional activities, that is, pathological habits of prolonged teeth clenching, grinding, and tapping [2].

The specificity of the anatomical structure of temporomandibular joints, and especially soft tissue structures, is conducive to the morphological consequences of excessive pressure exerted on the condyle, occurring during occlusal (biting, clenching the teeth) and non-occlusal parafunctions (nail biting, cheek biting, chewing on objects). A very precise system of reflex regulatory mechanisms, together with soft tissue proprioceptors, the impulse conduction system of the second and third branch of the trigeminal nerve, the activity of the sensory nucleus, and the participation of the reticular formation, thalamus, and cerebral cortex, and consequently the activity of the motor nucleus, as a response to previous impulses from the oral cavity, ensure efficient regulation of physiological conditions, but also inform about disorders occurring within all structures of the musculoskeletal system of the masticatory organ [1,2,3,4,5,6].

Generally, TMD symptoms fluctuate over time and correlate significantly with masticatory muscle tension, teeth clenching, grinding, and other parafunctional behaviors [1,2,6,7].

The aim of this article is to present cases of medical emergencies in a group of patients suffering from TMD and the therapeutic methods that are recommended, based on one’s own clinical experience and the literature studied.

## 2. Disc Displacement without Reduction

This is a condition in which the disease of disc displacement with reduction progresses into a form of disc displacement without reduction. The acoustic symptoms previously occurring within the temporomandibular joints (in the form of clicking or popping sounds during the movements of the mandible) disappear. This forward position of the discs causes elongation and damage to the posterior disc’s ligaments; simultaneously, the loss of flexibility causes the disc’s inability to return to its correct position relative to the condyle (the central occlusion) [4,5]. The reason for this condition is often the long-term occurrence of occlusal parafunctions with prolonged tension of the muscles responsible for lifting the jaw (raising the mandible), including the upper lateral pterygoid muscles, the attachment of which (about 30% of the fibers) is located within the anterior part of the intra-articular discs [1,6,7,8].

This condition is manifested by a sudden limitation of the jaw opening range (on average about 28 to 33 mm) and deviation of the mandible towards the joint where the disc is blocked. Predominantly, the disc is blocked in one of the temporomandibular joints, there is the limitation of lateral movement in the opposite direction to the blocked disc, and pain of varying severity (most often over 5 on the VAS scale) (Visual Analogue Scale) on a commonly used pain intensity scale [7], occurs in the preauricular area of the joint. It is crucial to unlock the displaced disc as soon as possible, considering the damage to the soft tissue structures of the temporomandibular joints [1,2,5].

Therapy and rehabilitation in the case of such dysfunctions should include an attempt to actively unblock the articular disc by manual manipulation of the mandible. Firstly, the mandible must be placed by the patient laterally, in the direction of the opposite side regarding the blocked disc; subsequently, the mouth must be opened to the full range by the practitioner while keeping the mandible in the lateral position. Such activities should be repeated several times during one session. This method is very effective when the patient consults the doctor shortly after the disc displacement without reduction occurs [1].

If this method fails to be effective, the manipulation should be performed, in the same manner as in the case of a mandibular dislocation. A thumb should be placed on lateral teeth located in the lower jaw, while the rest of the fingers should be wrapped around the mandible. The pressure ought to be applied downwards in the area of the molars; furthermore, the lower front edge of the mandible should be pulled upwards. After such manipulations and achieving a reduction in the articular disc, it is recommended to use an “immediate” anterior repositioning appliance, made of impression material (Figure 1), for a period of 2–3 weeks for 20 h a day, except for eating [1,6,7].

A follow-up appointment should be scheduled a week after the disc reduction (unblocking) procedure, which is equal to a week of the usage of the “immediate” appliance by the patient. Depending on the reported further symptoms, treatment is carried out in accordance with the rules applicable. The application of intraarticular injections with platelet-rich plasma and hyaluronic acid brings usually positive benefits [5,6,8,9,10], despite the opinion that these studies still need to be continued to clearly determine the positive effects of using platelet rich plasma in the rehabilitation of the temporomandibular joints [5].

## 3. Sudden and Acute Contraction of the Inferior Lateral Pterygoid Muscle—Myospasm

Uncontrolled and prolonged use of an anterior repositioning splint may lead to a complication in the form of a sudden, reversible contraction of the inferior lateral pterygoid muscle [9]. This condition should be differentiated from the inflammation of the pericardial tissues. It is manifested by the sudden inability to connect the teeth in the lateral section of the dental arch on the side of contraction, difficulties in opening the mouth wide, spontaneous pain, and tenderness on palpation in the area of the inferior lateral pterygoid muscle. The degree of contraction of this muscle may vary, yet the pain that occurs is most often referred to as excruciating [1,2,3,11].

Therapeutic procedures aim to relax the pterygoid muscle and return it to its regular length. When using the anterior repositioning splint, the mandible is moved anteriorly and the inferior lateral pterygoid muscles are stiff. The relaxation of these muscles is carried out by moving the lower jaw back under the pressure of the fingers placed in the area of the chin. The second method proposed by Wright E. and Klasser G. [1] is the above-described manipulation with the position of the thumb on the lateral teeth and the remaining fingers under the mandible, as well as intraoral compression therapy in the palate area behind the eighth upper teeth. If, as a result of the myospasm of the inferior lateral pterygoid muscle, the prolonged and uncontrolled use of the anterior repositioning splint takes place, a temporary break in the use of this appliance should be recommended.

## 4. Attack of Fibromyalgia, When the Disease Previously Undiagnosed

Fibromyalgia syndrome (FMS) is a little-known disease, classified as a group of diseases in the field of “functional” pain. It is characterized by chronic, generalized pain in muscles and joints, and the occurrence of the so-called tender (trigger) points, i.e., places on the human body which are excessively sensitive to pressure (from 11 to 18 places) (Figure 2). Many symptoms may resemble other musculo-articular disorders. The patient reports a state of constant fatigue and the fact that sleep does not bring the feeling of rest [12,13,14]. It most often affects people in the third and fourth decade of life. The symptoms are very bothersome: they significantly deteriorate the quality of life and hinder everyday functioning of the patient.

Fibromyalgia is associated with abnormal pain sensation of unknown etiology; moreover, central sensitization is possible—excessive excitability of pain-conducting structures in the spinal cord and brain. When suffering from fibromyalgia, the pain in muscles and bones is very distressing; it can be diffuse, chronic, piercing, throbbing, and prickly. Patients coping with fibromyalgia are characterized by having abnormal levels of substances involved in pain sensation (e.g., decreased serotonin levels) and struggling with a deep sleep disorder, which could explain fatigue and an inability to recover during the night. Moreover, genetic susceptibility, psychological factors (chronic stress, depression, mobbing, occupational crises), infectious diseases (Lyme disease, HIV), and autoimmune diseases (rheumatoid arthritis, lupus erythematosus, Hashimoto’s disease) are mentioned in the context of fibromyalgia [12,14].

Treatment is difficult and not always effective. In cases of fibromyalgia attacks affecting patients suffering from TMD, sedatives, relaxation exercises, and hot compresses in the area of the muscles of mastication should be used [13].

## 5. Hypertrophy of the Coronoid Process of the Mandible

This medical condition was discovered in 1853 by surgeon von Langenbeck [15]. It may occur unilaterally but most often occurs bilaterally; it concerns bone hypertrophy on three levels (Figure 3). This problem is more common among men than women. The etiology is unknown. Clinical symptoms are non-specific and may lead to misdiagnosis. Although this process takes many months, the reduction in the jaw opening range occurs suddenly and worsens. Due to this specificity of the sudden onset of a reduction in the jaw opening range, practitioners tend to ignore the correct diagnosis [1,13,14,15,16].

A particular type of hypertrophy of the mandibular coronoid process occurs in children’s organisms, especially in the cases of genetic diseases; in such a situation, a multi-stage surgical treatment combined with physiotherapy is necessary [17,18,19].

In diagnostics, clinical examinations, computed tomography, CBTB, and scintigraphy are used. Most often, hypertrophy takes a painless form. The differentiation concerns fibrous dysplasia, joint tumors, and hemifacial hyperplasia. The treatment offered in condylar hypertrophy is surgical exclusively (coronoidectomy) and followed by physiotherapy [1,18,19].

## 6. Acute Inflammation of the Temporomandibular Joints; Specific Inflammation

Acute arthritis can affect all joint structures, periarticular tissues, and the joint capsule. Most often it is associated with the aftermath of an injury, blood-borne infection in the course of infectious diseases, rheumatic disease, or by continuity from surrounding tissues: parotid glands (Figure 4), skin boil, jawbone, or the inner and outer ear. Diagnostics are based on body temperature tests, CRP and ESR levels. In the process of differential diagnosis, such factors should be considered: complex extraction of wisdom teeth, parotitis, otitis media, parotid and rumen abscess, osteitis of the ramus of mandible. Treatment includes mandible immobilization, administration of antibiotics, painkillers, and anti-inflammatory drugs [20,21].

## 7. Myofascial Pain Dysfunction Syndrome—MPDS

This syndrome is characterized by the sudden onset of pain in the masticatory muscles, head, and neck area on one side of the body. This indicates the interdisciplinary and complex nature of these ailments. In the etiology of many facial pain syndromes, the role of psychoemotional factors and depression is indicated. The possible neurogenic or vascular origin of this pain is emphasized. The prevalence of orofacial pains is assessed for at least 10% of the adult population and 50% of the elderly population [1,2,20].

In addition, the sensitivity of the tissues during the palpation of the muscles occurs, together with the excessive tension and frequent contractions of muscles associated with stress, patient anxiety, frequent parafunctional activity, or craniofacial injuries. This may be accompanied by acoustic symptoms within the temporomandibular joints. Furthermore, the comorbidity of MPDS and fibromyalgia is possible. The development of trigger points in the masticatory muscles is often observed. The diagnosis is based on a clinical examination and analysis of the reported symptoms and the elimination of the possibility of the dental origin of pain (pulpitis, increased dentin sensitivity, caries, periapical inflammation, and periodontopathies) [1,2,6,21].

## 8. Secondary Trigeminal Neuralgia

Contrary to the very characteristic symptoms of the primary trigeminal neuralgia (sudden, short-term, stabbing pain on one side of the face; additionally, very intense convulsions may occur, except for the bedtime), the secondary form is characterized by the pain of a different nature: constant muscle pain (which may become stronger with time) occurring suddenly, but lasting without interruption. It may be a consequence of many local causes (dental or sinus diseases, cysts, complications after the extraction of wisdom teeth) and general causes (diseases of the posterior cranial fossa, cerebellopontine angle tumors) [22,23,24,25].

If the character of such a disease is bilateral, tumors at the base of the brain or skull may be the cause. The symptoms intensify when the pathological areas are exposed to heat. The results of magnetic resonance imaging or computed tomography of focal lesions (aneurysms, tumors, osteophytes, focal demyelination in multiple sclerosis) are essential in the process of diagnosis of secondary neuralgia. Treatment for the secondary form involves the elimination of the primary lesion and the use of pharmacotherapy, including antiepileptic drugs (carbamazepine), which stabilize nerve impulse conduction. Moreover, the following procedures are performed: skin rhizotomy, electrophoresis, acupuncture, microvascular decompression, or stereotaxic radiosurgery [23,26].

## 9. Subluxation of the Temporomandibular Joint 

The subluxation of the temporomandibular joint can be caused by the trauma or flaccidity of the muscles responsible for lowering the jaw. When the patient is unable to close the mouth from the maximal open position, it means that the head of the mandible is in front of the articular tubercle. Subluxation is diagnosed when the patient is able to overcome the problem oneself, whereas luxation is diagnosed when the patient needs someone else’s help to return the condyle to the proper position within the joints [25]. The reasons for this condition may be found in: craniofacial trauma (most often in the chin area), removal of wisdom teeth, a sudden act of biting a hard object, congenital laxity of the soft tissues of the temporomandibular joints, and flaccidity of the muscles responsible for lowering the jaw. The rehabilitation of patients coping with polyarticular laxity and disorders connected with collagen structure, such as osteogenesis imperfecta, should be carried out under special supervision [1,6,27,28,29]. The consequence is the displacement of the condyle in front of the articular tubercle. In the case of unilateral subluxation, the lower jaw deviates to one side. A characteristic symptom is an inability to connect the jaws in the central occlusion position [30,31].

Therapeutic treatment includes adjustment and immobilization of the mandible for a period of 10 days with the use of a chin cap or intermaxillary traction. In addition, sclerotizing agents, designed to induce inflammatory reactions and secondary contraction of the joint capsule can be injected; moreover, acupuncture can be performed [1,2,6,27,32,33,34].

## 10. Injuries of the Temporomandibular Joints

Injuries of the temporomandibular joints are primarily contusions; their characteristic symptom is the occurrence of spontaneous pain (which intensifies during the movements of the mandible), joint swelling, and deviation when lowering the mandible. The above-mentioned symptoms may be accompanied by limitation of the jaw opening range, intraarticular hemorrhages, rupture of the joint capsule or intraarticular disc, condylar fracture, or chondromalacia of the articular surfaces. Usually, treatment includes immobilization of the mandible for a few days and then unloading the joint with the use of occlusive splints [29,30,31,32,33].

Table 1 contains the summary of the emergency conditions and their typical symptoms.

## 11. Conclusions

In conclusion, it should be emphasized that the treatment for the conditions discussed (emergencies) is very complex and difficult, requiring the use of many modern methods of basic and complementary treatment. Modern methods of treatment primarily include: the fight against pain sensation in masticatory muscles and joints, prosthetic procedures with the use of modern equipment, such as kinematic facebows and Arcus Digma axiography, occlusal splints, interarticular injections of hyaluronic acid, PRP, and botulinum toxin type A as a method of treatment for joint pain and relaxation of masticatory muscles. Very useful and helpful for patients are physiotherapeutic procedures, such as; kinesitherapy, sonophoresis, manual therapy, post-isometric muscle relaxation, and the use of a biostimulation laser [1,2,31,32].

It should be emphasized that the occurrence of each of the aforementioned cases is a difficult task for the doctor and requires additional diagnostics, often different from the current therapy and requiring taking into account the general health of the patient.

## Figures and Tables

**Figure 1 jcm-12-00298-f001:**
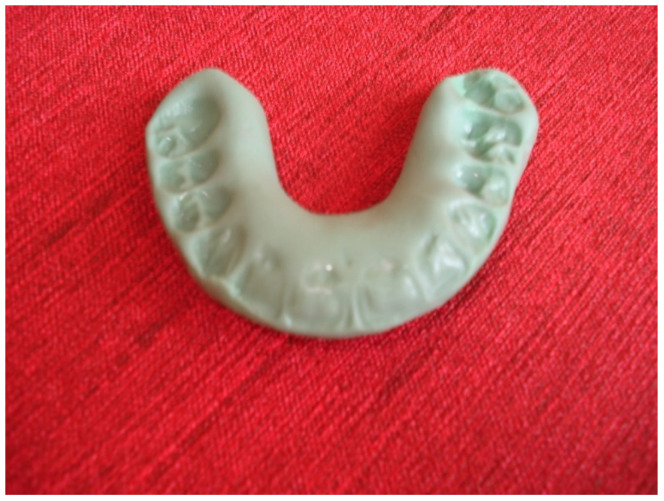
Temporary anterior repositioning appliance made of impression material.

**Figure 2 jcm-12-00298-f002:**
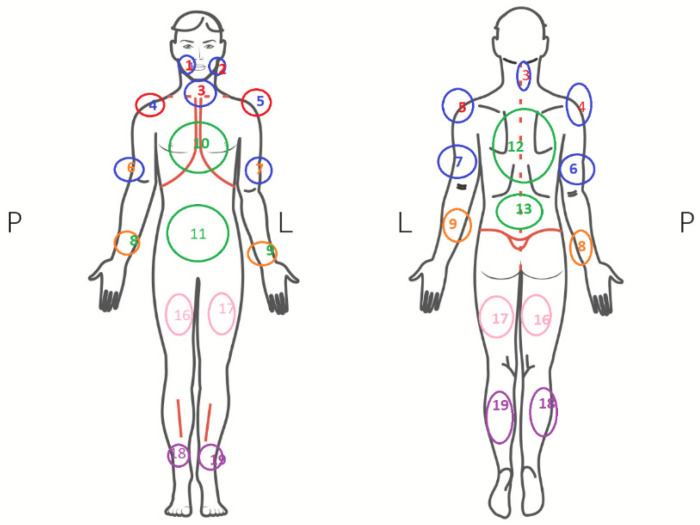
Tender points of the muscles in fibromyalgia [12].

**Figure 3 jcm-12-00298-f003:**
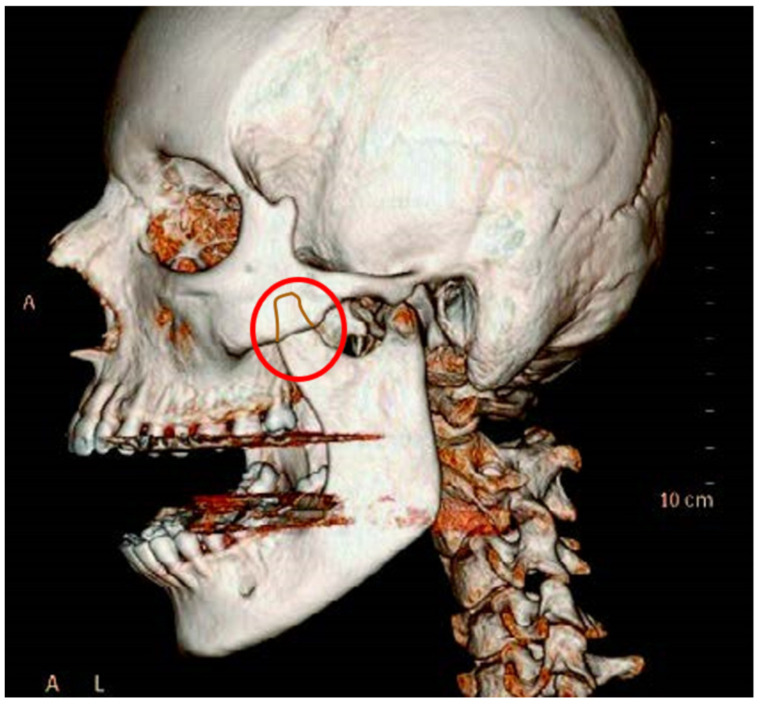
Pathological hypertrophy of the coronoid process of the mandible.

**Figure 4 jcm-12-00298-f004:**
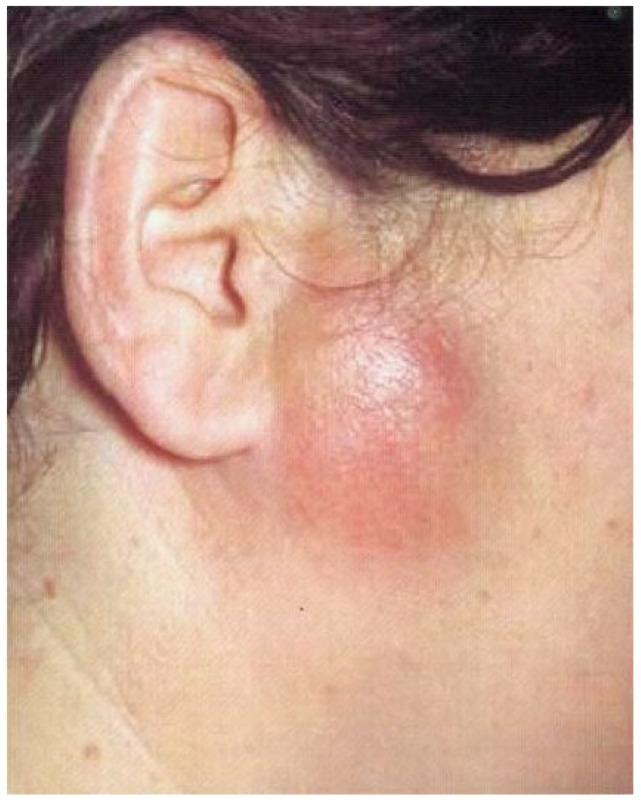
Parotid abscess can cause the inflammation of the joints by continuity from surrounding tissues.

**Table 1 jcm-12-00298-t001:** Summary of emergencies and their symptoms in the case of TMD patients.

	The Form of Emergencies in TMD	The Symptoms
1.	Disc displacement without reduction	a sudden limitation of the jaw opening range (on average about 28 to 33 mm), deviation of the mandible towards the joint where the disc is displaced and blocked
2.	Sudden and acute contraction of the inferior lateral pterygoid muscle—myospasm	a sudden inability to contact the teeth in the lateral section of the dental arch on the side of contraction, difficulties in opening the mouth wide, spontaneous pain, and tenderness on palpation in the area of the inferior lateral pterygoid muscle
3.	Attack of fibromyalgia, when the disease previously undiagnosed	the abnormal pain sensation of the masticatory muscle of unknown etiology, possible central sensitization, the pain in the skeletal muscles
4.	Hypertrophy of the coronoid process of the mandible	bone hypertrophy on three levels, the sudden onset of a reduction in the maximal interincisal distance range
5.	Acute inflammation of the temporomandibular joints; specific inflammation	diagnostics is based on body temperature tests, CRP and ESR levels, the differential diagnosis should include: complex extraction of wisdom teeth, parotitis, otitis media, parotid and rumen abscess, osteitis of the ramus of mandible
6.	Myofascial pain dysfunction syndrome—MPDS	a sudden onset of pain in the masticatory muscles, head, and neck area on one side of the body, the sensitivity of the tissues during palpation of the muscles occurs, together with the excessive tension and frequent contractions of muscles associated with stress, patient anxiety, frequent parafunctional activity, or craniofacial injuries
7.	Secondary trigeminal neuralgia	constant muscle pain (which may become stronger with time) occurring suddenly, but lasting without interruption
8.	Subluxation of the mandible	when the patient is unable to close the mouth from the maximal open position, it means that the head of the mandible is in front of the articular tubercle
9.	Injuries of the temporomandibular joints	occurrence of spontaneous pain, which intensifies during the movements of the mandible, joint swelling, and deviation when lowering the mandible. The above-mentioned symptoms may be accompanied by limitation of the jaw opening range, intraarticular haemorrhages, rupture of the joint capsule or intraarticular disc, condylar fracture, or chondromalacia of the articular surfaces

## Data Availability

Not applicable.

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
