# Peer review of "The Emergencies in the Group of Patients with Temporomandibular Disorders"

_jcm, 2022, doi:10.3390/jcm12010298_

Round 1

Reviewer 1 Report (New Reviewer)

This mini review is scientifically sound, although in large it needs extensive English editing as there are parts that are not clear to understand and it seems that the authors did not perform a sufficient revision. at some parts this language insufficiency can lead to misleading conclusions from the readers.

some examples that range from simple grammatical errors to totally ambiguous sentences:

in the first sentence from the intro:

 - Temporomandibular disorders (TMD) are an increasingly common disorder

- tooth clenching, grunding and other oraz parafunctional behav- 51 iors [1,2,6,7].

- which appeared before during the movements of the mandible

and so goes on...

-

beside the language insufficiencies, there are parts of the text that need to be clearer for the reader, as an example:

"In the most cases there is disc blocked in one of the joints, limitation of lateral 71 movement in the opposite direction to the blocked disc and joint pain in the preauricular 72 area, of varying severity, most often over 5 on the VAS scale."

 - what is the VAS scale? this needs to be mentioned and cited too.

- intraarticular injections with platelet-rich plasma and hyaluronic acid brings 96 very positive benefits [1,2,8].

such therapy is still controversial in many countries and the theraputic effects are still questionable in many of the TMD patients, this needs to be mentioned and better citations (meta-analysis or a systematic review) need to replace the current citations.

 as a final point, most of the authors citations ignore some of the most prominent works in the field and cite local papers or studies from their country of origin, which I find problematic for a review that is supposed to represent the state of the art and major/current literature (which is plenty!) in TMD studies. I highly advice the authors of rearranging their references and cite the proper literature in most positions where local citations are present.

Author Response

Dear Reviewer,

Thank you very much for the comments received in the review. All comments have been taken into account when correcting the text - the whole text has been corrected in terms of English grammar

  • misspelled words have been corrected, according to the comments - the abbreviation VAS has been explained and a citation has been added to this fragment
  • - a fragment of the text on the use of PRP in intra-articular injections and less enthusiastic opinions regarding this method and additional citations have been added.
  • None of the references quoted in this article come from local newspapers, but they are taken from the pub med or google scholar databases, and contemporary literature by authors expert in the discussed topic from around the world has not been omitted. Nor do I think that quoting literature from my country is a mistake.

Reviewer 2 Report (Previous Reviewer 2)

All of the issues have been addressed

Author Response

Dear Reviewer,
Thank you very much for accepting my changes made during the proofreading.
Thank you very much for your time and kindness

Round 2

Reviewer 1 Report (New Reviewer)

the manuscript is substantially improved and can be accepted when the editor deems it suitable.

This manuscript is a resubmission of an earlier submission. The following is a list of the peer review reports and author responses from that submission.

Round 1

Reviewer 1 Report

Authors need to provide informed consent for Figure 4

Authors need to include a para on  challenges with temporomandibular disorders

Author Response

Thank you very much for your comments. I reported the article as a preview, I don't know why there was an accidental change to the original article. I am very sorry for the trouble and confusion.

Figure 4 has been removed.

The article has been supplemented and corrected in accordance with everyone's guidelines of the Reviewers

Reviewer 2 Report

The manuscript is well written. However, I am not sure which category it should be in. It stated as Article which I expected to be an original article. However, there is no objectives as well as materials and methods.

Looks like a mini review but only 27 references. It should be about or more than 100 references for a review paper.

Is it a case study? It does not fall into this category as well.

I think the authors should revise the article into review article with more references.

Author Response

Thank you very much for your comments. I reported the article as an review one, I don't know why it finally appeared as an original article. I sincerely apologize for this problem. The article was supplemented and corrected in accordance with the guidelines of all Reviewers and submitted as a minireview.

Thank You very much

Reviewer 3 Report

the article presents a limited number of possible exacerbations of temporomandibular disorders, without making a complete systematization of them or bringing any original contribution in terms of diagnosis and or treatment of these conditions, like in Li DTS, Leung YY. Temporomandibular Disorders: Current Concepts and Controversies in Diagnosis and Management. Diagnostics (Basel). 2021 Mar 6;11(3):459.

Author Response

Thank you very much for your comments. I reported the article as an illustrative one, I don't know why it finally appeared as an original article. I sincerely apologize for the problem. The article was supplemented and corrected in accordance with the guidelines of all reviewers and submitted as a minireview.

A limited number of possible exacerbations was connected with our experiences.  The aim of the study was to point the problem of different diagnosis in TMD but not systematization of them. We describe special algorythm of unblocking disc and treatment during 3-5 weeks with special appliance, wchich we  belive are original contribution in terms of treatment of these conditions. In the literature we can find classifications but only few of publications with algorythm of treatment.

This summary of such emergencies may provide a valuable guide to the clinical management of such cases. That is original solution, which was not summarized in references before.

Reviewer 4 Report

Although it is an interesting topic, it lacks scientific rigour. It gives a brief description of the most frequent clinical entities related to temporomandibular disorders observed in a dental department. The authors do not provide data about the number of patients seen, their characteristics, personal background, and distribution according to the different TMJ pathologies.

Additionally, there are some editorial errors in the abstract and in the text, when referring to the two forms of trigeminal neuralgia. In point 2 of the text, referring to disc displacement without reduction, the reference to loss of flexibility of the posterior disc ligament is repeated.

Figure 4: The description in the main text does not match the legend, I do not understand why author mentioned parotid abscess appears because parotiditis is mentioned as a differential diagnosis.

Figure 5: Having a caption at the bottom of the figure, then a sentence in red letters expressing the same idea is redundant. It seems this information from the cited website has been copied and paste without having been adapted to the study.

Figure 6: The quality of the tomography is not good and there is misspelling.

Table: The sentence in the legend is repeated, in a different format, at the top of the table.

Table: Does not have the same numbering as the captions in the text. This would be fixed by not including the introduction in the text with the number 1, nor the conclusion.

Bibliography: There are more references in the text -29- than corresponding citations in the list -27-. Citations 15 and 16 are the same.

I would suggest to the authors to plan a well-designed original study on this topic with statistical analysis.

Author Response

Thank you very much for your comments. I submitted  the article as an review one, I don't know why it finally appeared as an original article. I sincerely apologize for this problem. The article was supplemented and corrected in accordance with the guidelines of all Reviewers and submitted as a minireview.

  1. The abstract of the article has been corrected
    2. Description of Fig. changed. 4-parotid gland abscess
    3. Figures 5 and 6 have been completely removed from the text of the manuscript
    4. Number of  subsection 1 and 11 of the introduction and conclusion chapter have been deleted. 5. The references have been completed and corrected
  2. Thank you very much for Your constructive and very helpful comments.